# Pretreatment of Copper Sulphide Ores Prior to Heap Leaching: A Review

**Andrés Neira** [1], **Diana Pizarro** [1], **Víctor Quezada** [1,*] and **Lilian Velásquez-Yévenes** [2]

1   Laboratorio de Investigación de Minerales Sulfurados, Departamento de Ingeniería Metalúrgica y Minas, Universidad Católica del Norte, Avenida Angamos 0610, Antofagasta 1270709, Chile; ans004@alumnos.ucn.cl (A.N.); diana.pizarro@alumnos.ucn.cl (D.P.)
2   Escuela de Ingeniería Civil de Minas, Facultad de Ingeniería, Universidad de Talca, Curicó 3340000, Chile; lilian.velasquez@utalca.cl
*   Correspondence: vquezada@ucn.cl

**Abstract:** Although the main cause of hydrometallurgical plant closures is the depletion of oxidized copper minerals reserves, the lack of new hydrometallurgy projects also contributes to these closures. One solution is to be able to process copper sulphide ores hydrometallurgically. However, it is widely known that sulphide copper ores—and chalcopyrite in particular—have very slow dissolution kinetics in traditional leaching systems. An alternative to improve the extraction of copper from sulphide ores is the use of a pretreatment process. Several investigations were developed evaluating the effects of pretreatment, mainly in the extraction of copper from chalcopyrite in chloride media. This study presents a review of various pretreatment methods prior to heap leaching to aid in the dissolution of copper from sulphide ores. Different variables of pretreatment that affect the extraction of copper were identified, including the type of salts used in agglomeration, curing time, and curing temperatures. Successful cases such as the implementation of the CuproChlor® process (use of calcium chloride), and various pilot studies using sodium chloride and temperature, show that pretreatment is an alternative that aids in the dissolution of copper from sulphide ores.

**Keywords:** copper sulphide ore; agglomeration; leaching; curing; pretreatment; chloride



## 1. Introduction

Chalcopyrite represents approximately 70% of the world's copper mineral reserves. This mineral is still one of the most refractory minerals when treated by hydrometallurgical methods. For this reason, since the 19th century, 70% of world production was generated through conventional processes of concentration by flotation followed by smelting [1]. In Chile, 40% of the copper produced is through smelters, and the rest is marketed as a concentrate for later smelting. Due to technological backwardness and lack of innovation, the concentration–smelting process requires high water consumption for the mineral concentration, and high energy consumption. It also causes environmental problems due to the continuous generation of air pollution and sulphur dioxide ($SO_2$) emissions [2,3]. Despite this, the projections made by the Chilean copper commission (Cochilco) [4] forecast an increase in the production of refined copper in this manner. This increase would reach an amount of 6.24 million tonnes by 2029, increasing the associated environmental problems. Cochilco (2019) [5] reported that the use of continental waters reached 12.45 $m^3/s$, where the activity in flotation plants uses 64% of this amount. Without a doubt, soon, the environmental regulations to protect the atmosphere, consumption, and contamination of the water will be much stricter. Therefore, the treatment of these minerals by this route will be restricted

The other 30% of world copper production is carried out mainly by processing oxidized copper ores by hydrometallurgical means. The depletion of these mineral resources is of great concern, causing the closure of various hydrometallurgical plants, which will

be less and less active in the country's productive matrix. Even though the main cause of the closure of the hydrometallurgical plants is the depletion of oxidized copper mineral resources, the lack of new hydrometallurgy projects also contributes to these closures. The projections made by Cochilco (2020) [6] reflect that by the year 2031, there will be an installed capacity in hydrometallurgical plants that will be unused to around 2500 kilotonnes. The heap leaching technique is the most widely used technique at the industrial level within the hydrometallurgical processes for oxidized copper ores. The widespread use of heap leaching is due to its simplicity, and the fact that it entails short residence times, where it is generally necessary to reduce the size of the ore in crushers without the need to go through milling [7–9].

Various investigations were carried out through hydrometallurgical processes to treat copper sulphide ores and search for new alternatives to feed idle hydrometallurgical plants in the future [10,11]. The main problem was the refractoriness of sulphide ores—especially chalcopyrite. Leaching of chalcopyrite was attempted via extremely fine particle sizes, involving high temperatures, and environmental pressures or higher, but unfortunately, none of these techniques could be carried out at the industrial level [12]. Other methods are based on bacterial leaching, where microorganisms allow the oxidization of ferrous ions to ferric ions, which are then capable of oxidizing the mineral [13]. Unfortunately, this methodology is very slow, and is only profitable for low-grade secondary sulphide ores.

In 2010, Nicol et al. published numerous scientific articles dealing with the slow dissolution kinetics of chalcopyrite in acid–chloride solutions and at moderate temperatures (20–35 °C). The laboratory-scale results of these researchers were promising, showing an improvement in copper extraction [14–16]. Some mining plants have used these acid–chloride solutions to treat sulphide ores, using the heap leaching technique with some modifications. The main improvements were to agglomerate the sulphide ore with acid–chloride solutions, with prolonged curing times as pretreatment, followed by heap leaching [9].

The pretreatment method prior to leaching was investigated in recent years, and the information available is scarce [17]. Pretreatment is conducted according to stages, such as agglomeration and acid curing, which are fundamental for improving leaching. Some of the benefits obtained by performing a correct pretreatment process include shorter leaching cycles, greater extraction of metals, lower operating costs, and lower acid consumption, which in some cases generates the use of simpler, cleaner technologies [17–19]. One of the pretreatments used is chemical, which shows favourable results, also being used in the treatment of gold minerals [18,20]. Regarding studies of pretreatment in agglomeration and curing, several studies carried out over the last 5 years stand out, such as [17,21–25]. Most of these studies proposed the addition of salts (both dissolved and solid), mainly for the incorporation of chloride into the system. The curing time or resting time is reported as one of the most important variables that aids in the dissolution of copper.

Although the most studied pretreatment process is agglomeration and acid curing, there are other proposals prior to these stages. The authors Moravvej et al. [26] studied the effects of microwave irradiation as a pretreatment for sulphides and oxidized copper ores via leaching tests in shaking flasks, to later carry out a conventional leaching process, obtaining a copper extraction of 6.05% for sulphide copper ores without pretreatment and 8.17% with pretreatment, while for copper oxide minerals, a copper extraction of 84.81% was obtained without pretreatment, and 93.74% with pretreatment, in addition to a decrease in the consumption of sulphuric acid of 28.8% for copper sulphides and 10.5% for copper oxides.

There are pretreatment proposals other than those of copper metallurgy. Such is the case of the study carried out by Qiu et al. [27], which consists of a cyanidation process with a pretreatment of adding pyrite to achieve greater extraction of silver. For this experiment, the authors carried out a pretreatment with pyrite as a reducing agent, achieving an increase in silver extraction of 43.92%. Another study by Mesa and Lapidus [28] proposed a pretreatment with sodium hydroxide at room temperature for the extraction of gold from refractory arsenopyrite. The authors varied the use of a sodium hydroxide concentration,

the concentration of solids, and treatment time, obtaining a gold extraction of 29% without performing a pretreatment (thiosulphate leaching) process. With the use of pretreatment, a gold extraction of 81% was achieved, demonstrating the effectiveness of the process. Other research conducted by Chen et al. [29] proposed a pretreatment with $O_2$, $H_2$, and CO, for a sample of autocatalyst for the extraction of rhodium (Rh). A 56% extraction of Rh without performing any pretreatment was achieved, which increased to 82% with pretreatment.

An extensive review of the literature directly and indirectly related to the use of pretreatments before leaching of copper sulphide ores is presented in this paper. This study identified a gap in the literature, mainly due to the wide number of investigations focused on the leaching of sulphide ores, and the lack of research on previous stages or pretreatments, such as agglomeration and curing. For this reason, this investigation demonstrated that pretreatments are a key stage that leads to an increase in the dissolution of copper sulphide ores in heap leaching processes.

## 2. General Aspects

### 2.1. The Current Situation of Hydrometallurgy in Chile

The first references to hydrometallurgical processes indicate that they were used prior to the 16th century, but it was not until the early 20th century that they had a different approach from the one that had been used to treat copper ores. The process of heap leaching was used on a large scale in Chile, where copper oxide ores were treated with sulphuric acid, and copper sulphides were solubilized with the help of ferric ions, which acted as an oxidizing agent. In Chile, to obtain copper from leaching solutions, electrowinning (EW) was used (which is still in use today), as opposed to precipitation through the use of scraping iron [30].

A challenge faced by hydrometallurgical activity in Chile is the closure of its plants (heap leaching), such as the case of El Salvador (Codelco), announced in 2013. This closure was due to the lack of reserves (oxidized minerals) and high costs. However, this did not transpire, as different alternatives were sought to keep their operations running [31]. According to [1], due to the lack of mineral resources, some copper mines would close their leaching operations—among them are Mantoverde (2023), El Abra (2023), Mantos Blancos (2024), Spence (2024), Lomas Bayas (2024), Cerro Colorado (2026), Gabriela Mistral (2025), Sagasca (2025), Tres Valles (2026), Pampa Camarones (2026) (all of these plants through heap leaching), and Los Bronces (2027) (dump leaching), as well as some reduction in the production at Codelco, such as Chuquicamata and Radomiro Tomic (heap leaching). On the other hand, according to [32], the plants that would not close their operations only represent 12% of the cathodes that will be produced by the year 2027, which are the following:

- Encuentro óxidos: This mining site will make use of the Tesoro Plant (heap leaching) when its mineral reserves run out. It should be noted that this mining company is adjacent to Centinela Oxides [1].
- Escondida: According to the projections, their oxide resources will be exhausted by 2025, but they would still have the run-of-mine (ROM), which corresponds to the low-grade minerals (dump leaching) that were extracted and sent to a large heap. These extracted minerals were not previously crushed, and will continue to supply the solvent extraction and electrowinning plants [1].
- Chuquicamata óxidos: The run-of-mine of this mining company could deliver solutions to the solvent extraction and electrowinning plants of Chuquicamata. However, the cathode production will be affected, which is estimated to decrease compared to that obtained in 2014 through the hydrometallurgical pathway [1].

Another project that would save the Chuquicamata mining operations is the retreatment of leaching residues and artificial resources (dump leaching) belonging to leachable materials that contain copper. These come from a stock generated from the heap mineral treatment plant (PTMP), the solvent extraction plant, and the oxide electrowinning plant—all obtainable resources to continue operations. It is under consideration to increase the

treated tonnes from 30,000 t/day to 45,000 t/day, for which 70% of the material from Mina Sur and 30% from Chuquicamata would be used, providing a total of 145 million tonnes. With this, there is an expected life extension of nine years [33].

Hydrometallurgy production in Chile will suffer a decrease in its participation in total copper production, given that in 2019 it had a 27.3% participation (1,580,000 tonnes), while in 2031 an 8.1% participation is expected (578,000 tonnes) [6]. If one looks at the operations of the hydrometallurgical plants at present and in the future, it can be seen that there are 31 plants actively operating, while in the future, in 2030, 19 will remain operational—of which 5 belong to Enami, 8 are from large mining companies, and 6 from medium-sized mining companies [34].

Among some of the projects that are under consideration in maintaining the hydrometallurgy route, Centinela has one; this consists of small projects, so that its oxide plant remains operational (2026–2040). Other active companies will undertake projects to continue viability in the coming years, such as: Planta Nora (2020–2035), Diego de Almagro óxidos (2021–2031), Rayrock—which will resume (2021–2035)—Producción óxidos (2021–2031), Playa Verde (2023–2030), and Marimaca (2025–2040) [34].

### 2.2. General Aspects of Copper Sulphide Leaching

Copper sulphide leaching is highly dependent on the redox conditions of the system and the addition of oxidizing agents. In some cases, temperature and pressure conditions are needed to favour the process. The most commonly used oxidizing agents are oxygen, ferric ions ($Fe^{3+}$), nitric acid ($HNO_3$), concentrated sulphuric acid ($H_2SO_4$), and cupric ions ($Cu^{2+}$) [35].

The relative kinetics of different copper minerals depend on the copper mineral species: carbonates, sulphates, and chlorides have very fast leaching kinetics at room temperature; cupric oxides and silicates have fast kinetics, but require higher acidity; native copper, cuprous oxides, and some silicates and complex oxides with manganese have moderate kinetics, and require an oxidant; simple and complex sulphides have slow/very slow kinetics, and require an oxidant [36].

There are different leaching media that are used to leach chalcopyrite; among the most common are sulphates, chlorides, nitrates, ammonia, and bacteria [37]. Leaching with ammonia has the peculiarity that huge amounts of ammonia need to be used, due to the generation of ammonium sulphate (($NH_4)_2SO_4$), which must later be decomposed in order to obtain the ammonia reagent [38]; this process is carried out at a temperature between 75 and 80 °C, with rapid stirring, in the presence of oxygen, and at a low pressure [39]. According to [40], Reactions (1) and (2) are those that govern the process:

$$2CuFeS_2 + 8.5O_2 + 12NH_3 + 2H_2O = 2Cu(NH_3)4SO_4 + 2(NH_4)_2SO_4 + Fe_2O_3 \qquad (1)$$

$$(NH_3)_2SO_4 + Ca(OH)_2 = 2NH_3 + CaSO_4 \cdot 2H_2O \qquad (2)$$

Nitric acid, having the characteristics of an effective oxidizing agent, is able to successfully dissolve various sulphide minerals with acceptable kinetics. However, this reagent has a high cost; thus, the reagent has a negative economic impact, and hinders the viability of the process [41]. The use of nitric acid as an oxidizing agent is more effective when it is in the presence of $NO^+$. Adding $NO^{2-}$ accelerates the generation of $NO^+$, thereby allowing the oxidation of the sulphide minerals at low temperatures, and forming elemental sulphur [41–43]. According to [43], Reactions (3) and (4), are the possible reactions that govern the process:

$$3MeS + 2NO_3^- + 8H^+ = 3Me^{2+} + 3S^0 + 2NO + 4H_2O \qquad (3)$$

$$MeS + 2NO_3^- + 4H^+ = Me^{2+} + S^0 + 2NO_2 + 2H_2O \qquad (4)$$

An unconventional alternative is the use of chloride media, offering significant advantages, such as presenting high metal solubility and better leaching rates. One of the advantages of working with chloride ions is that the chemical activity of the proton is increased. The activity coefficient for chloride salts is generally significantly higher than the values for the corresponding sulphate salts. In chloride solutions, the hydrometallurgical processes in chloride media take into account the oxidizing capacity of $Fe^{3+}$ and $Cu^{2+}$ ions for the oxidation of sulphide ores to elemental sulphur, as well as the high stability of the metal chloride complexes in solution [44–47]. Studies suggest that both ions participate in the oxidation reactions; However, the leaching capacity of cupric ions is greater than that of ferric ions, since cupric ions tend to regenerate more easily in the presence of oxygen, while ferric ions form strong complexes with chloride ions. Additionally, high concentrations of chlorides in solution allow $Cu^+$ ions to be thermodynamically stable from a thermodynamic point of view; $Cu^{2+}$ ions are available for dissolution reactions to occur.

Several studies were conducted searching for a medium that best favours the dissolution of chalcopyrite. It was determined that the dissolution kinetics in chloride media are the highest compared to those of sulphate media. This can be explained as being due to the formation of a chloride–copper complex, which has the characteristic of being able to stabilize $Cu^+$ ions in strong complexes, therefore allowing the cupric ions to act as oxidants of chalcopyrite [48,49].

The bioleaching of minerals is another treatment available for the exploitation of copper sulphide ores. This process of leaching has the advantages of not generating a high cost of water usage, nor generating large operating and production costs, in addition to emitting low levels of pollutants to the environment. Another benefit of bioleaching is that it extends the use of solvent extraction and electrowinning plants [12,50]. The first steps of this technology at a commercial level were at Minera Pudahuel in Chile. With the passing of time, new improvements in operations based on this option emerged, revealing the profitability and functionality of this process [51].

The main role of microorganisms in the process of mineral bioleaching is to oxidize ferric ions and sulphur compounds. The authors of [52] proposed the following reactions, where Reaction (5) corresponds to the oxidation of the ferrous ions by the action of iron-oxidizing microorganisms, and Reaction (6) to the oxidation of sulphur compounds, which become sulphates due to the interaction of sulphur-oxidizing microorganisms:

$$2FeSO_4 + 0.5O_2 + H_2SO_4 = Fe_2(SO_4)_3 + H_2O \tag{5}$$

$$S^0 + 1.5O_2 + H_2O = H_2SO_4 \tag{6}$$

The microorganisms present in bioleaching can be classified by using temperature gradients. The most prominent microorganisms are moderate thermophiles with optimal temperatures from 40 to 55 °C, extreme thermophiles with optimal temperatures higher than 55 °C and, finally, mesophilic microorganisms, which have an optimal temperature below 40 °C [53].

It should be noted that using thermophilic microorganisms presents an advantage when bioleaching, as an increase in temperature generates an increase in the speed of the oxidation reactions of minerals [54].

## 3. Pretreatments Methods

### 3.1. Mechanical Operations

Comminution is one of the first pretreatments performed on copper minerals before dissolution, whether they are oxides or sulphides. This is one of the most important steps within mining operations, consisting of reducing the size of the rocks by mechanically fracturing and grinding the minerals to obtain smaller fragments.

The objective of comminution is to free the mineral of interest from the gangue, increase the surface area of the particles, and produce particles of adequate sizes for the

subsequent processes, such as leaching, and transportation of the mineral. This operation is carried out continuously in different stages. The first is the primary crush, preceding the secondary/tertiary crush and, finally, leading to the grinding stage [55]. It should be noted that traditional heap leaching processes do not require a milling stage.

However, the comminution or crushing operations present a challenge—the expenditure of energy. Regardless of whether the crushing is carried out under or above ground, this operation continues to consume the most energy [56]. As reported by [56], the mineral crushing operation has an approximate range of 2–3% of energy consumption worldwide.

A given solution to reduce energy consumption is by the use of high-pressure grinding rollers (HPGR), which have significantly lower energy consumption. This type of equipment is used to achieve the release of diamonds, for the preparation of iron ore, for granulation, and—in recent times—in mining treatments of hard rocks, such as gold, platinum, and copper [57].

As reported by [58], the use of hPGR technology in copper leaching using columns provides a greater release of sulphides at the grain boundaries, greater fracturing of the rock matrix, better accessibility of scattered, fine-grade sulphides located in fractures and, finally, an increase in the leaching kinetics with respect to the conventional comminution in use. In certain cases, hPGR grinding can increase copper extraction by 2–10%, in a grinding circuit.

Another treatment studied to reduce energy consumption in crushing operations is the use of microwaves, which was studied for several decades [59]. Likewise, the use of microwaves on minerals produces mineralogical changes, which are favourable for mineral processing—especially for lateritic mineral processing.

The main objective of the use of microwaves is to generate cracks through thermal stresses. In order to fracture the mineral, it is subjected to differential thermal expansion, where the minerals that are within the rock are exposed to cycles of increasing temperatures and subsequent cooling, resulting in the fracture of the mineral. Fracturing also occurs due to the production of tensions within the mineral particles [60].

Depending on the type of rock treated, the reaction to microwave irradiation will be different; this is especially the case due to the dielectric properties of minerals, generating this heating behaviour differential [61].

The main cause of dielectric materials heating up is the variation in the electric field, being the dominant mechanism in microwave heating. Another method by which a microwave thermal process can be performed is ionic conduction [62].

As reported by [63], the use of microwaves was studied during the 1980s in the United States, testing the dielectric and low-power heating properties for common ore minerals. Studies showed that sulphates, micas, aluminosilicates, and carbonates exhibited minor heating. On the other hand, sulphides and metallic oxides were easily heated when exposed to microwave energy.

As previously mentioned, microwave irradiation depends on both the chemical and physical properties of the mineral to be treated. In the case of absorbent minerals, the microwave manages to enter the interior directly, whereas for the transmitters, the microwaves are reflected, not being able to enter the interior of the mineral [26].

Once the microwave pretreatment is complete, it can generate a lower consumption of grinding energy, in addition to achieving a better extraction of valuable metals [64]. However, energy greater than 10 kW h/t is generally required for a low power density, which means that no energy savings are generated during the crushing operations. Furthermore, residence time is greater than 1 s, which prevents the implementation of an operation with thousands of tonnes per hour, as is required in the mining industry [63]. As reported by [9], a high treatment capacity would be necessary to work with several generators in parallel, which would not generate an economic benefit with respect to a conventional process. On the other hand, if minerals with high-value products or lower tonnage were processed, the use of microwaves could be an alternative to traditional processes [9].

*3.2. Agglomeration and Curing*

One of the most widely used pretreatments to increase copper extraction in the heap leaching process is agglomeration and curing, which is performed prior to the heap leaching stage [19]. This is done in the first instance to avoid permeability problems in the heap. However, it was also determined that this stage can better dissolve minerals. Additionally, proper handling of the agglomerate and curing process can indicate success for the overall heap leach operation. However, there is no fixed process for bonding and curing—rather, it is based on experience [65].

The agglomeration process consists of smaller particles adhering to larger ones, forming a glomer, which results in the formation of openings in the leach heaps. Therefore, it is essential to obtain good permeability for gases and liquids of the agglomerated minerals [66]. On the other hand, if the mineral is not subject to an agglomeration process, there is the possibility that small particles will mix together with the leaching solution, covering the flow channels and pores, causing the permeability to decrease, which generates low dissolution of the mineral [65].

The agglomeration process is one of the methods to obtain good recovery in a heap leach. If inappropriate agglomeration takes place, it may be one of the main causes of low extraction [67]. Inappropriate agglomeration is due to an erroneous quantity of moisture added to the agglomeration process, which means that the fine mineral particles do not adhere to the large mineral particles, due to either the lack or excess of moisture. Another consequence of performing this process incorrectly is that the glomers present poor mechanical resistance, causing them to break when being transported to the leaching heap, leading to a segregation of the particle size in the heap [68]. On the other hand, the benefits of performing this process correctly are higher dissolution rates; that is, the leaching cycles take less time, and there is both an improvement to the conditions and a better structure of the heap. This is because channelling is minimized, improving the permeability and availability of reagents [17,19]. When the mineral to be treated has large amounts of clay, when large amounts of fine minerals are produced in the crushing process, or when a mineral is crushed to a size of 0.75 inches (19 mm) or finer, an agglomeration process will be required [67].

On the other hand, as a complement to agglomeration, the use of binders was investigated. As the particles are not united with a great force, this causes the glomers to disintegrate, causing a migration of the fines. Binders are a potential solution to this problem, since they help the formation of a more stable, strong, and disintegration-resistant glomer. For a binder to be effective, it must withstand the acidic environment that is present in heap leach operations, as well as having a strong affinity for mineral particle surfaces. The binder to be used should not affect the leaching chemistry or subsequent processes [69,70].

It is for this reason that binders must be able to create chemical bonds in order to obtain a stable glomer. Different studies were conducted regarding the use of lime, weeds, and wood fibres, but the results were not satisfactory; the resulting glomers when using these binders completely disintegrated when immersed in water for a couple of hours [71]. The use of Portland cement as a binder for gold and silver provides a better resistance to the formed glomer; this is because calcium silicate hydrates are formed during the curing process; However, these glomers partially or entirely disintegrate when they dry when using less than 50 kg/t of cement [66].

The choice of binder must be based on the mineral to be treated and the conditions of the desired product [72]. These can be classified into different types, such as polymeric, organic, or inorganic. In the case of a precious metal heap leaching under alkaline conditions, Portland cement is used, while for acidic conditions, diluted or concentrated sulphuric acid is widely used [67].

According to [73], organic binders such as modified cellulose and lignin were chosen because they are difficult to degrade. Modified cellulose is a hydrophilic component, which allows it to absorb water, allowing for retention of a part of the leaching solution

that is in contact with it, and allowing it to remain attached to the surface of the mineral. Other binders—such as gelatine, agar, sodium carboxymethyl cellulose, gums, and starch—proved to be inefficient under acidic conditions. Inorganic binders such as sodium silicate, which were chosen due to their reaction with an acidic medium, produced a silica gel that can act as a binder. Other inorganic binders—such as calcium sulphate, iron(II) sulphate, and sodium tripolyphosphate—were also tested under acidic conditions, with poor results. Polymeric binders were shown to have the ability to resist degradation by acid solutions; they have the ability to bind to hydrogen ions, which are adsorbed on the surface of minerals [73].

Instead, curing is intended for the mineral to interact with the leaching solution early on, causing the mineral to form sulphate (copper sulphate). The generation of this sulphate is a benefit for the leaching process, due to the high solubility of this product. Another effect of curing is that it can reduce the passage of silica in the leaching process, thanks to the fact that it solubilizes iron, which generates ferric ions, which in turn dissolve sulphides, avoiding the formation of colloidal silica [17]. As the acid-curing process proceeds, some of the components that have already dissolved react again and precipitate, thus causing a better bond with the mineral [74].

Acid curing generates the dehydration of aluminium silicate minerals by partially eliminating a monolayer of hydroxide that covers these silicates, causing the surface to become insoluble and hydrophobic in aqueous solutions. In addition, it homogenizes the distribution of the acid in the mineral bed and generates greater porosity in the bed, improving permeability [75]. The curing time varies depending on the mineral being treated, which can be in short periods of time—such as 8–24 h—or in long periods of 1–15 days, or even longer.

The curing effect is beneficial when performing a pretreatment for copper sulphide ores [76]. According to [77], which carried out a pretreatment process with sodium chloride, sodium nitrate, and sulphuric acid, and a curing time of 3 days, the authors report that a 64.7% copper extraction was obtained—in contrast to the test without pretreatment, where 26.8% copper extraction was obtained with the same operating conditions. In studies evaluating the effect of pretreatment, [65] stands out. In this investigation, tests were carried out with long curing times of greater than 48 days, and the addition of 35 kg/t of sodium chloride directly to the agglomerate; the authors note that the dissolution of copper sulphides (mainly chalcopyrite) was enhanced. Another investigation [21] focused on the study of the effect of temperature on curing time, where the authors proposed to evaluate the curing time at 50 °C in copper sulphide minerals—mainly of chalcopyrite and bornite—obtaining an improvement in extraction. Research by [78] investigated the use of sodium chloride, sulphuric acid, and ferrous sulphate in the agglomeration and curing stage. The authors of [78] report that the use of these reagents in the agglomeration and curing stage allows for an improvement in the extraction of copper compared to an agglomeration without the use of the ferrous sulphate reagent, using 0.6, 0.53, and 0.5 kg/50 kg of mineral, respectively, allowing it to cure for 14 days at a temperature of 32.9 °C. The effect of curing on the leaching of exotic copper oxidized minerals was also studied [79], evaluating the effects of the curing time and the sodium chloride concentration; the authors concluded that both variables influence the responses; furthermore, long curing times favour the reduction of $MnO_2$, which increased the dissolution of copper in the system.

### 3.2.1. Agglomeration with Calcium Chloride

CuproChlor® is a process created for the leaching of copper sulphides, and initially it was intended to be used in secondary copper sulphides. This process provides a chloride medium for leaching generated by the addition of calcium chloride in the agglomeration stage. The process consists of four stages: agglomeration, resting or curing, leaching with recirculation solution, and washing with a refining solution.

In 2004, this process was patented by Minera Michilla, by the authors Abraham Backit Gutierrez, Jaime Rauld Faine, Raul Montealegre Jullian, and Freddy Aroca Alfaro. The

process consists of an agglomeration stage and a curing time, during which sulphuric acid, water, and calcium chloride are added, and reacts to form calcium sulphate or gypsum [80], as can be seen in Reaction (7):

$$CaCl_2 + H_2SO_4 + H_2O = CaSO_4 \cdot 2H_2O + h^+ + Cl \tag{7}$$

The generation of $CaSO_4 \cdot 2H_2O$ allows it to act as a solid bridge for agglomeration. The addition of sodium chloride also helps the stability of the glomers and, therefore, of the bed; its hydrodynamic properties are improved, such as its liquid and gas permeability, its hydraulic conductivity and, finally, its porosity. These properties are related to the kinetics and equilibrium of oxygen transport, from the gas to the liquid phase. In addition, this produces a change in the solutions, which go from a sulphate medium to a chloride medium, generating an improvement in the kinetics of the reactions in the presence of the cupric ions [81]. In the curing stage, a large part of copper and other species are solubilized. The dissolution mechanics of the CuproChlor® process for copper sulphide ores is based on the leaching of sulphides in chloride media. Thus, the ferric ions present in the agglomerate stage, generated due to the interaction of sulphide minerals with sulphuric acid, react by leaching the copper sulphide minerals. The minerals that react are mainly covellite and chalcocite, as observed in Reactions (8) and (9), respectively [80]:

Chalcocite:

$$Cu_2S + 2Fe^{3+} = Cu^{2+} + CuS + 2Fe^{2+} \tag{8}$$

Covellite:

$$CuS + 2Fe^{3+} = Cu^{2+} + S^0 + 2Fe^{2+} \tag{9}$$

A higher concentration of chlorides allows for the formation of oxidizing conditions in order to obtain a rapid solubilisation of the copper sulphides during acid curing, and therefore gain an improvement in the extraction of copper during leaching. The oxidation–reduction mechanism is based on Cl and Cu ions, since these allow for the formation of chlorocuprics and chlorocuproses, which are complex ions that interact with ferric and ferrous ions [41].

Later, during leaching, the solutions are required to contain iron, copper, and chlorides—especially the latter. The regeneration of the ferric ions occurs thanks to the cupric ions, which come from the leaching of copper minerals. Due to the presence of oxygen, the formation of ferric ions occurs, resulting in the oxidation of cuprous ions. This is a self-catalytic process, and the reaction is continuous until the sulphuric acid or oxygen are depleted [82].

To eliminate possible copper precipitates, acid must be present in the reactions, and is thus added to the recirculating intermediate leaching solution. Furthermore, this acid allows oxidation reactions to continue. A greater focus should be placed on the washing stage of the organic loading before it is discharged, since the increase in chlorides in the solution can be transferred to the electrowinning stage, generating problems in the process [41].

Adding calcium chloride in the agglomeration stage improves the hydrodynamic properties, as previously mentioned. Thanks to these improvements, the passage of oxygen to the heap results in a more efficient irrigation stage. To verify this, different experiments were carried out on a semi-industrial scale (1000 tonnes) evaluating the effect of CuproChlor® [81], where copper extraction results of 86–96% were obtained. Of the total tests carried out, 81% of them reached 90% or greater copper extraction—four of which were between 86 and 90%.

The working conditions under which Minera Michilla operated were: 30 kg/t of sulphuric acid and 12 kg/t of calcium chloride in agglomerate, 90 g/L of chloride concentration in leaching, a leaching time of 110 days, a heap height of 2.5 m, an irrigation rate of 0.32 L/min/m$^2$, and a total copper extraction of 90% [80].

The CuproChlor® process, unlike other copper sulphide leaching methods, has several advantages: One of these advantages is the short heap leaching times, which range from 100 to 110 days [41]. According to the study carried out by [83], bioleaching has a leaching time of approximately 167 days in column tests, and a time of 270 days in the case of a pilot plant. On the other hand, since the washing stage works correctly, this means that the large amounts of chlorides do not degrade the organic reagents, nor are there are problems in the quality of the cathodes obtained. Good permeability is also obtained—both liquid and gaseous. The process has suitable stability, and can form heaps of up to 6 m high, without suffering segregation problems; it can work with both fresh and sea water, and/or in the presence of ions that prevent the proliferation of bacteria [41], in addition to being able to work with a temperature lower than that required by bacterial leaching—since it works at a temperature of 25 °C [1], while the bacterial processes require a temperature that varies between 40 and 55 °C, or greater than 55 °C, depending on the microorganism used [50]. As the CuproChlor® process is 100% chemical, it provides ease in working conditions—unlike bacterial leaching, which requires special care; it can also work with clay or fine contents, which would not be suitable in processes where bacteria are used [41].

### 3.2.2. The Use of Other Salts

In recent years, studies were conducted on pretreatment with various salts, and how these would affect the dissolution of copper. One such study [22] investigated the use of an acid–nitrate–chloride medium. A mineral with a chemical composition of 0.70% total Cu, 0.04% soluble Cu, 5.65% total Fe, and an acid consumption of 33.5 kg/t of minerals was used. The most abundant copper mineral in the sample was chalcopyrite, representing 84% of the total copper.

For this study, the following variables were analysed: temperature (25 and 45 °C); the addition of sodium nitrate (11.7 and 23.3 kg/t); curing time (20 and 30 days); and the addition of sodium chloride (2.1 kg/t from seawater and 19.8 kg/t added in solid form). To carry out the study, 10 samples were prepared, which were pretreated with sodium nitrate and sodium chloride in different amounts. The highest copper extraction was 58.6%, with the addition of 23.3 kg/t of $NaNO_3$ and 19.8 kg/t of NaCl, and after 30 days of curing at 45 °C prior to leaching. Finally, the authors proposed Reactions (10–13) that govern the curing process in an acid–chloride–nitrate medium [22]:

For $CuFeS_2$ (Reaction (10)):

$$2CuFeS_2 + 10H_2SO_4 + 10NaNO_3 + 4NaCl = 2CuCl_2 + Fe_2(SO_4)_3 + 10NO_2 + 4S + 10H_2O + 7Na_2SO_4 \tag{10}$$

For CuS (Reaction (11)):

$$3CuS + 4H_2SO_4 + 2NaNO_3 + 6NaCl = 3CuCl_2 + 2NO + 3S + 4H_2O + 4Na_2SO_4 \tag{11}$$

For $Cu_5FeS_4$ (Reaction (12)):

$$Cu_6FeS_4 + 12H_2SO_4 + 12NaNO_3 + 10NaCl = 5CuCl_2 + FeSO_4 + 12NO_2 + 4S + 12H_2O + 11Na_2SO_4 \tag{12}$$

For $FeS_2$ (Reaction (13)):

$$3FeS_2 + 4H_2SO_4 + 2NaNO_3 + 6NaCl = 3FeCl_2 + 2NO + 6S + 4H_2O + 4Na_2SO_4 \tag{13}$$

The study performed by [24] investigated the effects of an agglomerate with sulphuric acid, sodium chloride, and potassium nitrate—these being the parameters and the response variable of the extraction of copper. To conduct the tests, a chalcopyrite mineral was used, with 28.5% Cu, 22.8% Fe, and 29.7% S. After agglomeration, it was left to cure for 0, 5, 10, and 15 days; after completing the curing time, the authors proceeded to wash, filter, and measure the metal concentrations in the filtrate.

According to the authors, in the pretreatment, the effect of the application of potassium nitrate was minimal, as 9% of the copper extraction was associated with chalcanthite, which is soluble in water, and another 4% corresponded to covellite. Therefore, a solubility greater than 13% guaranteed dissolution of the chalcopyrite in the pretreatment process, which was not achieved under these conditions.

From the statistical data obtained, the curing time parameter most influenced the results, with 54.66%; sulphuric acid had a moderate contribution of 35.75%, while potassium nitrate had a low contribution of 0.61%. For the case of a pretreatment with sodium chloride, the curing time was the parameter that had the most influence on the extraction of copper, with 56.36%; sodium chloride had a moderate contribution of 23.09%, while sulphuric acid had an extremely low contribution of 1.78%.

Another study [21] focused on the effects of a sodium chloride pretreatment prior to leaching of a copper sulphide ore. Here, a copper sulphide was used, which contained 1.21% chalcopyrite and 0.54% bornite—these being the minerals with the highest contribution of copper. The pretreatment process was carried out with 10 g of the mineral, to which 20 kg/t of $H_2SO_4$ was added, along with 65 kg/t of sea water, and the addition of NaCl, which varied depending on the condition to be used. The mineral samples were homogenized to avoid loss to evaporation, and were left in a covered Petri dish. These plates were placed in a muffle so that the temperature did not drop below 50 °C, where they were left for the necessary time, while those at 20 °C were left in the laboratory with air conditioning to maintain temperatures. After the curing time, the samples were leached in an Erlenmeyer flask. The highest copper extraction was with a chloride concentration of 90 kg/t, with a curing time of 40 days at a repose temperature of 50 °C. As the chloride concentration increased, the copper extraction increased, as did the temperature and curing time.

Comparing the CuproChlor® process with the experiment conducted in [21], a variation of 5.14% was presented, representing the differences between the two processes. Conducting this experiment as a pilot showed that it can be an alternative treatment. However, the use of 90 kg/t of sodium chloride can present a problem in downstream processes, such as SX or electrowinning. One of the problems generated by chlorides is that the $Cl^-$ ion oxidizes, thus forming chloride gas, causing pitting. This problem consists in the corrosion of the cathodic surface that is not in direct contact with the rich electrolyte. Another problem that can arise from exposure to chlorides is the trapping of impurities—such as lead—on the cathode surface. An alternative to avoid these problems is to perform a wash such as the one used in the CuproChlor® process. Another important factor is the greater use of sodium chloride compared to the use of calcium chloride in the CuproChlor® process, which may represent a greater operating cost.

Effect of the Addition of Chloride

According to the study carried out by [65], incorporating sodium chloride into the agglomerate leads to an increase in copper extraction. To confirm this, the authors compared the effect that extraction would have with and without the addition of chloride. However, by increasing the amount of chloride from 20 to 70 kg/t, the extraction did not undergo a significant improvement. This coincides with the findings of [21,48,49], where better results were obtained with chloride additions of 50 and 90 kg/t. This may be due, as suggested by [75], to the fact that the pretreatment carried out with chloride and acid allows for the reactive agents to be distributed more homogeneously in the mineral, thus generating the dissolution reaction earlier. On the other hand, there is the formation of soluble species with a greater solid–liquid interaction, in addition to a greater range of porosity; in this way, moisture is retained in the pores. Moreover, the addition of chlorides caused an increase in the extraction of copper in the tests conducted, which is consistent with the study conducted by [84].

The study conducted by [21] presented an extraction of 92.86%, unlike the 63% obtained in the study performed by [22] after the leaching process; this difference of 29.86% is likely due to the greater use of sodium chloride in the pretreatment process, which presents a difference of 70.2 kg/t between both experiments, since there was not a big difference between the curing time and the curing temperature—which were 10 and 5, respectively.

Effect of Nitrate Addition

The addition of sodium nitrate in the pretreatment process resulted in a positive improvement in the tests performed by [22,24]. One of the possible reasons for this is that nitrates are strong oxidizing agents to decompose copper sulphides; this leads to the fact that when nitrates are added, it results in a greater quantity of oxidizing ions to leach copper sulphides [85,86]. A study conducted by [87] obtained a copper extraction of 92% using nitrates with ferric chloride. It was concluded that this was due to the fact that mixing high concentrations of sodium chloride and sodium nitrate in an acid medium is more effective, which was confirmed by the study conducted by [22]. In addition, the authors of [43] note that chalcopyrite does not react unless it has an oxidizing agent in a sulphuric acid system, which is consistent with the studies mentioned above. However, in the case of potassium nitrate, this may be due to the non-addition of chlorides, as this was not the purpose of the experiment. When comparing the copper extraction results of the experiments performed by [22,24], a copper extraction difference of 35.94% in the pretreatment stage was observed. This difference is due to the variables studied by each author, which were curing temperatures (these being 45 and 25 °C, respectively) and curing times (which were 30 and 15 days, respectively). A greater focus was given to the addition of sodium nitrate by [22], which was 21.3 kg/t. Checking the effectiveness of the addition of nitrates in the pretreatment stage, together with the aforementioned, can explain this variation in copper extraction between both experiments. It should be noted that the addition of chlorides was 19.8 kg/t and 25 kg/t, respectively.

Effect of Curing Time

Increasing the curing time favours greater copper extraction. Moreover, in the statistical analysis carried out by [24], the curing time was the most effective in increasing copper extraction. This could be due to the fact that, as the curing time is longer, there is more time for the acid and chlorides to react with the treated mineral, or there is longer exposure to a large ionic charge. This ionic charge is caused by sulphuric acid, sodium chloride, and sodium nitrate. In addition, the humidity of the sample must be such that it allows the reactions to occur more quickly [21]. This is consistent with the studies performed by [23], where the authors confirm that the curing time increases the extraction of copper from sulphide minerals, and [65], which notes that implementing prolonged curing times generates a benefit to the heap leaching operations that treat copper sulphides; this is due to the fact that this shortens the leaching time, which leads to an improvement in the management of the solution/water and decreases the irrigation requirements.

Effect of Temperature on Curing

By increasing the temperature in the curing process, a better dissolution of copper is obtained. According to the authors of [78], with a higher temperature, dissolution reactions occur faster, because less energy is needed to break the molecular bonds. In the study conducted by [78], the effects of 32.9 °C and 14.5 °C with a 14-day curing time were compared. Once the pretreatment was finished, the mineral was leached in columns, achieving a copper extraction of 78.9% and 5.9%, respectively. Another benefit that temperature provides in curing is that having a high temperature requires a shorter curing time, which is of benefit as the formation of copper sulphate occurs. The evaluation of the temperature, as part of the pretreatment, was investigated by [21]; the authors demonstrated that a maximum copper extraction of 93% was obtained when the minerals (mainly chalcopyrite) were treated with a pretreatment of 90 kg of Cl$^-$/t of mineral, 40 days of curing at 50 °C, and then the pretreated sample was leached at 45 °C in shake flasks. Although an increase in temperature in the curing stage aids in the dissolution of copper, it is also possible to identify successful cases at room temperature. The study performed by [88] indicates the benefit of curing time for chalcocite/covellite minerals in leach columns. Minerals agglomerated with sulphuric acid, chloride ions, and a long cure time have been shown to improve the dissolution rate of a secondary copper sulphide

ore at room temperature. The authors note that it is possible to obtain a 72% extraction of Cu when the mineral is agglomerated and cured for 50 days, without the need to increase the temperature in the curing stage. This is mainly due to mineralogy, considering that chalcocite is the easiest copper sulphide to dissolve—a situation very different from chalcopyrite [89].

## 4. Conclusions

Due to the depletion of oxidized copper ores, hydrometallurgical plants are running out of ore to process. Therefore, by dissolving the copper sulphide ores through heap leaching, it could be possible to produce copper in a sustainable way. An alternative to achieve this is by strengthening the agglomeration and curing stages prior to heap leaching.

Both physical and chemical pretreatment processes are important in the hydrometallurgical treatment of copper sulphide ores. The agglomerate and those cured present a greater importance in the treatment of copper sulphide ores. The CuproChlor® pretreatment process (calcium chloride addition) stands out, which produces a heap leaching time of 110 days, with copper extractions that vary between 86 and 96% at 25 °C.

In the reviewed studies, the addition of chlorides is favourable for the pretreatment process, increasing the extraction of copper. In the experiment performed by Cerda et al. [21], copper extractions ranging from 63.7% to 92.9% were achieved with additions of 90 kg/t of $Cl^-$, while in the investigation conducted by hernández et al. [22], an increase of 15.1% was achieved when the addition of chloride increased from 2.1 kg/t to 19.8 kg/t.

Adding sodium nitrate in the pretreatment process resulted in a positive improvement in the tests performed by hernández et al. [22]. By varying the addition of sodium nitrate from 11.7 to 23.3 kg/t, an increase in copper extraction of up to 13.7% was obtained. However, the tests conducted by Quezada et al. [24] using potassium nitrate did not achieve an increase in copper extraction with the addition of 10 kg/t of potassium nitrate.

In most of the studies reviewed, the most influential variable turns out to be the curing time. In the study conducted by Quezada et al. [24] it was determined that, according to the variables used, 55% of the contribution was due to the curing time, according to the ANOVA analysis.

Increasing the temperature during the curing process improves copper extraction. According to the research conducted by Cerda et al. [21], an increase in copper extraction of up to 23.5% was obtained when varying the temperature from 20 °C to 50 °C, while in the study conducted by hernández et al. [22], an increase in copper extraction of 8% was obtained by increasing the temperature in curing from 25 °C to 45 °C.

There are differences between studies at the laboratory and at the industrial scale, the main one being particle size. The particle size in the studies analysed at the laboratory scale varies from 0.0098 cm to 0.79 cm, compared to 1.91 cm used at the industrial level (heap leaching). Another variable that generates differences is the curing temperature, considering that on an industrial scale the leaching of minerals is at room temperature. These differences are the major limitations in the scaling of the various studies.

**Author Contributions:** Conceptualization, V.Q. and L.V.-Y.; methodology, A.N., D.P. and V.Q.; validation, V.Q. and L.V.-Y.; formal analysis, A.N., D.P., V.Q. and L.V.-Y.; investigation, A.N. and D.P.; resources, V.Q. and L.V.-Y.; data curation, A.N., D.P. and V.Q.; writing—original draft preparation, A.N. and D.P.; writing—review and editing, A.N., D.P., V.Q. and L.V.-Y.; visualization, A.N. and D.P.; supervision, A.N., D.P., V.Q. and L.V.-Y.; project administration, V.Q. and L.V.-Y.; funding acquisition, V.Q. and L.V.-Y. All authors have read and agreed to the published version of the manuscript.

**Funding:** This research received no external funding.

**Informed Consent Statement:** Not applicable.

**Conflicts of Interest:** The authors declare that they have no conflict of interest.

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
