# Peer review of "Pretreatment of Copper Sulphide Ores Prior to Heap Leaching: A Review"

_metals, doi:10.3390/met11071067_

Round 1
Reviewer 1 Report
This reviewer feels that the article is poorly presented and the following revisions needs to be addressed.
1) 11.6% Cu production in 2029 – this should be an estimated production.
2) Please report your data accurately – The authors mentioned flotation based Cu production 70%, pyrometallurgy 25%. If we add the given hydromet fraction, the total would be more than 100%.
3) Lines 71-77: What is the leaching methodology (tank leaching, heap or column leaching) to obtain the given extraction efficiencies?
4) Research gap is not explicitly highlighted in the literature. Without a research gap this study is rather superfluous.
5) Section 2.1 – The authors used the term hydrometallurgical operations without properly defining it. Tank leaching, heap leaching and dump leaching (for Cu overburden) belong to hydrometallurgical operations that produce Cu and it would be better to separate the operations clearly to give very clear perspective to the readers. This could be useful: https://www.sciencedirect.com/science/article/pii/S0892687518302760
6) In summary, the usage of the term “hydrometallugy” is too broad in this manuscript. It would be better to provide an introduction to different percolation leaching methods, including examples from the Chilean mining industry.
7) In heap leaching context, gas sparging from bottom of the heap is also considered as a possible strategy to increase primary sulphide leaching efficiency.
Author Response
Dear reviewer.
We really appreciate your reviews. Your reviews contributed to improving our document. We appreciate your time and dedication.
1) 11.6% Cu production in 2029 – this should be an estimated production.
- This section of the Introduction has been updated. This section was rewritten.
2) Please report your data accurately – The authors mentioned flotation based Cu production 70%, pyrometallurgy 25%. If we add the given hydromet fraction, the total would be more than 100%.
- The information refers to the treatment (mine production) of copper sulphide ores. This section was rewritten. This information was edited according to figures from the Chilean Copper Commission (Cochilco).
3) Lines 71-77: What is the leaching methodology (tank leaching, heap or column leaching) to obtain the given extraction efficiencies?
- The methodology used was leaching tests in shaking flasks. The study (Moravvej et al., 2018) was cited considering the novel pretreatment proposed by the author. The study is carried out on a laboratory scale but with interesting results.
4) Research gap is not explicitly highlighted in the literature. Without a research gap this study is rather superfluous.
- Pretreatment has been neglected and is an alternative that favors heap leaching. The current situation in Chile is complex for leaching, due to the depletion of oxidized minerals. Then, the future feeding of hydrometallurgical plants (Leaching) will be chalcopyrite, which is refractory. Also the copper grades decrease. An alternative to improve leaching is pretreatment. And this document presents the novelty of offering multiple pretreatment alternatives.
5) Section 2.1 – The authors used the term hydrometallurgical operations without properly defining it. Tank leaching, heap leaching and dump leaching (for Cu overburden) belong to hydrometallurgical operations that produce Cu and it would be better to separate the operations clearly to give very clear perspective to the readers. This could be useful: https://www.sciencedirect.com/science/article/pii/S0892687518302760
- The document focuses primarily on heap leach treatment. The concept of hydrometallurgical operation refers to a plant that includes leaching, solvent extraction and electrowinning. This definition has been more emphatically defined (If it is Heap Leaching or Dump leaching (includes ROM methodology). We really appreciate this review and the document as an example.
6) In summary, the usage of the term “hydrometallugy” is too broad in this manuscript. It would be better to provide an introduction to different percolation leaching methods, including examples from the Chilean mining industry.
- A better definition regarding heap leaching was incorporated in the Introduction. Mining companies of the past, current and future projects are also mentioned.
7) In heap leaching context, gas sparging from bottom of the heap is also considered as a possible strategy to increase primary sulphide leaching efficiency.
- You are right. It is a more focused alternative to leaching, but not as a pretreatment (before).

Reviewer 2 Report
Since this is a review paper, the authors are entitled to as many references as they want and this is granted. But, the title mentions "heap leaching" and the authors do not elaborate in the text on this technology. Is pretreatment followed by heap leaching? Further, why heap leaching and the sulfide copper concentrates can be leached in autoclaves - - which is not mentioned at all in the text. The authors mention in the abstract that ".. the lack of new hydrometallurgical projects contribute to this closure". The manuscript is long and tedious and is really not understood what the authors want.
Author Response
Dear reviewer.
We really appreciate your reviews. Your reviews contributed to improving our document. We appreciate your time and dedication.
The paper is focused on pretreatment (agglomeration and curing) before leaching. This paper does not refer to the leaching of copper concentrates, not even in an autoclave. The paper refers to the agglomeration and curing of copper sulphide minerals at an industrial level and alternatives at a laboratory scale. Hydrometallurgical plants have always been used to process oxides, but these are over. Causing the closure of these hydrometallurgical plants. One solution is to increase the number of hydrometallurgical projects (leaching) by improving the dissolution of copper sulphide minerals (refractories). And this document presents the novelty of offering multiple pretreatment alternatives.
The manuscript was corrected and sections that did not contribute to the document were removed. The introduction was rewritten. Now the paper is easier to read. Finally, minor revisions regarding English grammar were edited.

Round 2
Reviewer 1 Report
Firstly, prepare a common "response to reviewers" letter including comments raised by the other reviewers (if any), otherwise this reviewer do not know the comments raised by the other reviewers and the authors addressed those comments or not.
Line 1 – mineral reserves
Line 55 – heap leaching, not “the heap leaching”
Lines 55-59 – Add recent references for these sentences
Line 60-69 – This is also written without any references
Line 73-77 – Add relevant references
Still research question is not given in the manuscript (what is the gap in the literature, why we need to address it) and without it, this manuscript would be superfluous.
Tons should be tonnes
Lines 194-200 – in Chile?
Line 292 - Fe3+ y Cu2+??
Use past tense for sentence such as “The authors [46] proposed….” – throughout the manuscript.
Cupro-Chlor® process – Introduce this in the text for the readers.
Conclusions – discuss the application of your work in the Chilean mining industry, especially copper. That should be the objective of a review paper.
Author Response
Dear reviewer,
We really appreciate all the time and reviews. The document improved. Thank you so much.
We respond to your reviews
1) Firstly, prepare a common "response to reviewers" letter including comments raised by the other reviewers (if any), otherwise this reviewer do not know the comments raised by the other reviewers and the authors addressed those comments or not.
- The comment of the other reviewer, in this second round, was:
I am glad that the authors have cut their manuscript and removed unnecessary parts. I hope that the final manuscript can be rightly produced.
2) Line 1 – mineral reserves
- Do you mean line 12? If so, the line was modified by “copper minerals reserves”.
3) Line 55 – heap leaching, not “the heap leaching”
- Done
4) Lines 55-59 – Add recent references for these sentences
- The following references were added
Van Staden, P. J.; Petersen, J. Towards fundamentally based heap leaching scale-up. Miner. Eng. 2021, 168, 106915, doi:10.1016/j.mineng.2021.106915.
Petersen, J.; Dixon, D. G. The dynamics of chalcocite heap bioleaching. In Proceedings of the TMS Fall Extraction and Processing Conference; 2003; Vol. 1, pp. 351–364.
Ghorbani, Y.; Franzidis, J. P.; Petersen, J. Heap leaching technology - Current State, innovations, and future directions: A review. Miner. Process. Extr. Metall. Rev. 2016, 37, 73–119, doi:10.1080/08827508.2015.1115990.
5) Line 60-69 – This is also written without any references
- The following references were added
Viñals, J.; Roca, A.; Hernández, M. C.; Benavente, O. Topochemical transformation of enargite into copper oxide by hypochlorite leaching. Hydrometallurgy 2003, 68, 183–193, doi:10.1016/S0304-386X(02)00200-1.
Córdoba, E. M.; Muñoz, J. A.; Blázquez, M. L.; González, F.; Ballester, A. Leaching of chalcopyrite with ferric ion. Part I: General aspects. Hydrometallurgy 2008, 93, 81–87, doi:10.1016/j.hydromet.2008.04.015.
Watling, H. R. The bioleaching of sulphide minerals with emphasis on copper sulphides - A review. Hydrometallurgy 2006, 84, 81–108, doi:10.1016/j.hydromet.2006.05.001.
Peng, T.; Chen, L.; Wang, J.; Miao, J.; Shen, L.; Yu, R.; Gu, G.; Qiu, G.; Zeng, W. Dissolution and passivation of chalcopyrite during bioleaching by acidithiobacillus ferrivorans at low temperature. Minerals 2019, 9, 4–13, doi:10.3390/min9060332.
6) Line 73-77 – Add relevant references
- The following reference was added
Ghorbani, Y.; Franzidis, J. P.; Petersen, J. Heap leaching technology - Current State, innovations, and future directions: A review. Miner. Process. Extr. Metall. Rev. 2016, 37, 73–119, doi:10.1080/08827508.2015.1115990.
7) Still research question is not given in the manuscript (what is the gap in the literature, why we need to address it) and without it, this manuscript would be superfluous.
- The following text was added in Introduction
An extensive review of the literature, directly and indirectly related to the use of pretreatments before leaching of copper sulphide ores, is presented in this paper. This study identified a gap in the literature, mainly, due to the wide number of investigations focused on leaching of sulphide ores and the lack of researching on previous stages or pretreatments, such as agglomeration and curing. For this reason, this investigation demonstrated that pretreatments are a key stage that leads to increase the dissolution of copper sulphide ores in heap leaching processes.
8) Tons should be tonnes
- We really appreciate this review. It was made throughout the document.
9) Lines 194-200 – in Chile?
- Yes, in Chile. It was specified in the text now.
10) Line 292 - Fe3+ y Cu2+??
- It was modified. Fe3+ and Cu2+.
11) Use past tense for sentence such as “The authors [46] proposed….” – throughout the manuscript.
- Was checked.
12) Cupro-Chlor® process – Introduce this in the text for the readers.
- At the beginning of section 3.2.1 an introduction to the process was carried out.
13) Conclusions – discuss the application of your work in the Chilean mining industry, especially copper. That should be the objective of a review paper.
- The following conclusion was added. Please consider that the other conclusions are in a comparative context for heap leaching (industrial scale).
Due to the depletion of oxidized copper ores, hydrometallurgical plants are running out of ore to process. Therefore, by dissolving the copper sulphide ores through heap leaching, it would be possible to produce copper in a sustainable way. An alternative to achieve this is by strengthening the agglomeration and curing stages, prior to heap leaching.

Reviewer 2 Report
I am glad that the authors have cut their manuscript and removed unnecessary parts. I hope that the final manuscript can be rightly produced.
Author Response
Dear reviewer,
We really appreciate all the time and reviews. The document improved. Thank you so much.
I am glad that the authors have cut their manuscript and removed unnecessary parts. I hope that the final manuscript can be rightly produced.

Round 3
Reviewer 1 Report
Address the minor corrections below:
Lines 95 and 97: Use alternative words instead of “face”
Line 97: It was tried – it implies what?
Line 160: proposed
Line 594: “sulphides, and initially it was….”
Line 600: The sentence starts with “Which” – Use “The process consists of an…”
Author Response
Dear reviewer,
We really appreciate all the time and reviews. The document improved. Thank you so much.
We respond to your reviews
Lines 95 and 97: Use alternative words instead of “face”
- Lines 95 and 97 were rewritten.
Line 97: It was tried – it implies what?
- Line 97 was rewritten
Line 160: proposed
- Was corrected
Line 594: “sulphides, and initially it was….”
- Was corrected
Line 600: The sentence starts with “Which” – Use “The process consists of an…”
- Was corrected
